# Cerium-Doped Iron Oxide Nanorod Arrays for Photoelectrochemical Water Splitting

**DOI:** 10.3390/molecules27249050

**Published:** 2022-12-19

**Authors:** Hai-Peng Zhao, Mei-Ling Zhu, Hao-Yan Shi, Qian-Qian Zhou, Rui Chen, Shi-Wei Lin, Mei-Hong Tong, Ming-Hao Ji, Xia Jiang, Chen-Xing Liao, Yan-Xin Chen, Can-Zhong Lu

**Affiliations:** 1School of Rare Earth, Jiangxi University of Science and Technology, Ganzhou 341000, China; 2CAS Key Laboratory of Design and Assembly of Functional Nanostructures, and Fujian Provincial Key Laboratory of Nanomaterials, Fujian Institute of Research on the Structure of Matter, Chinese Academy of Sciences, Fuzhou 350002, China; 3Xiamen Key Laboratory of Rare Earth Photoelectric Functional Materials, Xiamen Institute of Rare-Earth Materials, Haixi Institutes, Chinese Academy of Sciences, Xiamen 361021, China; 4School of Rare Earths, University of Science and Technology of China, Hefei 230026, China; 5College of Chemical Science, University of Chinese Academy of Sciences, Beijing 100049, China; 6Fujian Science & Technology Innovation Laboratory for Optoelectronic Information of China, Fuzhou 350108, China

**Keywords:** cerium-doped, Fe_2_O_3_ nanorod arrays, photoelectrochemical water splitting

## Abstract

In this work, a simple one-step hydrothermal method was employed to prepare the Ce-doped Fe_2_O_3_ ordered nanorod arrays (CFT). The Ce doping successfully narrowed the band gap of Fe_2_O_3_, which improved the visible light absorption performance. In addition, with the help of Ce doping, the recombination of electron/hole pairs was significantly inhibited. The external voltage will make the performance of the Ce-doped sample better. Therefore, the Ce-doped Fe_2_O_3_ has reached superior photoelectrochemical (PEC) performance with a high photocurrent density of 1.47 mA/cm^2^ at 1.6 V vs. RHE (Reversible Hydrogen Electrode), which is 7.3 times higher than that of pristine Fe_2_O_3_ nanorod arrays (FT). The Hydrogen (H_2_) production from PEC water splitting of Fe_2_O_3_ was highly improved by Ce doping to achieve an evolution rate of 21 μmol/cm^2^/h.

## 1. Introduction

Renewable energy for the sustainable development of the whole society is gaining momentum globally [1]. Hydrogen energy as a clean energy carrier is a non-polluting new energy source with high energy density [2]. The common approaches of electrolysis of water and steam reforming methane for H_2_ production either consume large amounts of electricity or fossil energy [3,4]. The photoelectrochemical (PEC) water splitting using solar light has been considered the most promising and environmentally friendly technique for H_2_ production [5,6,7,8,9,10]. In the PEC system, photoanodes absorb sunlight and split water directly into H_2_ and O_2_. Thus, the efficiency and stability of H_2_ production from PEC water splitting highly depend on the photoanodes materials [11,12,13,14].

Metal oxide semiconductors (MOS) have long been developed as photo-active materials for PEC water splitting owing to multiple advantages of superior stability, non-toxicity, and low cost [15,16]. Among various MOS, hematite (α-Fe_2_O_3_) as photoanodes has outperformed virtues of abundance, chemical stability, and suitable electronic structure. The α-Fe_2_O_3_ has a small band gap (2.2 eV) with the absorption of approximately 40% of the incident sunlight [17], which gives it a high theoretical solar-to-H_2_ (STH) efficiency. However, hematite has some fatal disadvantages that affect its photocatalytic performance; for instance, low electrical conductivity, high electron-hole complexation rate [12]; hole diffusion length is only 2–4 nm, which is relatively small compared to TiO_2_ (100 μm) and WO_3_ (100–150 nm) [18], low electron mobility (10^−1^V^−1^s^−1^) [19], and short lifetime of the excited state [20]. These have seriously limited the application of hematite for H_2_ production from PEC water splitting.

Many strategies, such as non-metal sensitization [21,22], rare earth elements doping [23,24,25,26], heterojunction construction [27,28], and surface modification [29,30,31,32], have been employed to enhance the PEC performance of Fe_2_O_3_. In addition, the ordered array structures can transform 1D nanorods into 3D spatial structures. They can also be used in various ways by preparing corresponding photoanodes depending on the shape of the substrate [33]. Seungkyu Kim et al. [34] have achieved twice the hydrogen yield of pure ZnO nanowires by constructing CeO_2_ nanoparticle-modified Ce-doped ZnO nanowire photoanodes; P. Senthilkumar et al. [35] prepared Ce Doped BaTiO_3_ Nanoassemblies achieving a hydrogen yield of 22.5 μmol h^−1^ cm^−2^, however pure BaTiO_3_ Nanoassemblies had a hydrogen production of 13.5 μmol h^−1^ cm^−2^; Yu Tan et al. [36] synthesized Ce/Ce_2_O/CeO_2_/TiO_2_ nanotube arrays could achieve hydrogen production of 5 mL h^−1^ cm^−2^ (≈223.21 μmol h^−1^ cm^−2^). The specific test conditions and experimental values are given in Appendix A.

Herein, the Ce-doped Fe_2_O_3_ nanorod arrays were prepared by a simple one-step hydrothermal method. The morphology and structure of Fe_2_O_3_ (FT) and Ce-doped Fe_2_O_3_ (CFT) samples were observed by X-ray diffraction (XRD), Scanning Electron Microscope (SEM), and Transmission Electron Microscopy (TEM). The PEC performances of Fe_2_O_3_ and Ce-doped Fe_2_O_3_ were comprehensively investigated. The optical properties and photocatalytic activity of Ce-doped Fe_2_O_3_ were remarkably improved. The Ce doping effectively promotes the charge transfer and separation of photogenerated electron/hole pairs in Fe_2_O_3_. The applied voltage also plays an important role in exerting the function of Ce doping in Fe_2_O_3_. Therefore, the Ce-doped samples realized high photocurrent density and incident photon-to-current efficiency (IPCE) values. The PEC performance for H_2_ production of Fe_2_O_3_ was highly improved by Ce doping.

## 2. Results

### 2.1. Structure and Morphology

XRD results confirm the hematite phase of as-prepared samples, FT, 1CFT, 5CFT, and 10CFT, as shown in Figure 1. The peaks at 24.13°, 33.12°, 33.61°, and 49.52° correspond to (012) (104) (110) (024) planes of the hematite phase in Fe_2_O_3_ nanorods (PDF#89-0596), respectively, and peaks at 35.08°, 38.40°, 40.16°, and 52.99° originate from the titanium substrate (PDF#89-2762). The rutile phase was also observed, indicating the oxidation of titanium TiO_2_ during the annealing process. The titanium substrate was successfully loaded with Fe_2_O_3_ nanorods films. The peak at 24.13°, which corresponds to the (012) crystal plane of Fe_2_O_3_, slightly shifted to a low angle depending on the cerium content. The Ce doping causes the lattice expansion in the Fe_2_O_3_.

SEM observed the morphology of the samples FT, 1CFT, 5CFT, and 10CFT, as shown in Figure 2a–d. The homogenous rods with an average length of 350 nm are perpendicular to the substrate, indicating that the morphology of nanorods was not affected by Ce doping. The Fe_2_O_3_ nanorods arrays consisted of 5–30 nm nanopores with mesoporous structures, respectively. The TEM measurements were also conducted to study the microstructure of Fe_2_O_3_ nanorods further. The hollow pore structures of nanorods in samples FT and 5CFT are directly observed, as shown in Figure 2e,g, respectively. The HRTEM of FT and 5CFT in Figure 2f,h show the lattice stripes with a specular spacing of 0.268 nm and 0.274 nm for the (104) crystal plane of Fe_2_O_3_, respectively. The Ce doping has led to a larger lattice spacing of the (104) crystal plane in Fe_2_O_3_, which is consistent with the XRD results. The HAADF image in Figure 2i and EDS mapping results in Figure 2j–m confirm the homogeneous distribution of Ce elements in Fe_2_O_3_ nanorods in the samples of 5CFT. Therefore, the Ce ions are successfully doped in Fe_2_O_3_ nanorods without affecting the morphology of Fe_2_O_3_ nanorods.

### 2.2. Chemical Composition and Surface Information

For obtaining the chemical composition and surface information of Fe_2_O_3_ and Ce-doped Fe_2_O_3_, XPS measurements were conducted. Except for the pristine FT, the Ce-doped samples of 1CFT, 5CFT, and 10CFT consist of C, O, Fe, and Ce, as shown in Figure 3a. As the Ce content increases, the position of the valence band shifts to the lower energy, as shown in Appendix A. The valence band position of 5CFT decreases vastly to 1.29 eV from the 1.65 eV in pristine FT. In the spectrum of Fe 2p, there are two peaks at 711.2 eV and 724.5 eV corresponding to Fe 2p_3/2_ and Fe 2p_1/2_, respectively, as shown in Figure 3b. In addition, there is a satellite peak at 718 eV, which is the signature of Fe (III) [37]. The Fe 2p peaks in the spectra of the Ce-doped samples shifted significantly towards lower binding energy, which is caused by the interaction between the Fe^3+^ and Ce^3+^ or Ce^4+^ substances [38]. In the spectrum of O 1s, the central peak at around 530 eV in FT corresponds to the lattice oxygen O_L_, as shown in Figure 3d. The shifting of O_L_ towards lower binding energy after Ce doping is due to the weak electronegativity of Ce ions.

In the Ce 3d spectra, the distinctive peaks of elemental Ce are seen at 889.1 eV, 894.2 eV, 898.5 eV, 903.2 eV, 907.1 eV, 916.5 eV in Ce doped samples as shown in Figure 3c. The peaks at 889.1 eV, 898.5 eV, 907.1 eV, and 916.5 eV originated from the Ce^4+^ [39]. Precisely, the former two peaks correspond to the Ce 3d_5/2_ orbital and the latter to the Ce 3d_3/2_ orbital. The signal at 894.5 eV originates from the spin–orbit coupling for the Ce 3d_5/2_ and Ce 3d_3/2_ states [40]. The spectra of Ti 2p show a standard bimodal curve for titanium dioxide, as shown in Appendix A. Therefore, the Ce ions are successfully doped in Fe_2_O_3,_ and the valance band position of Fe_2_O_3_ was reduced significantly.

## 3. Discussion

### 3.1. Photoelectrochemical Performance

The UV-Vis absorption spectra also investigated the optical properties of Fe_2_O_3_ photoanodes. The Ce doping specifically improved the light absorption performance of the Fe_2_O_3_ photoanode in the region of 800 to 620 nm, as shown in Appendix A. The band gaps of pristine FT and Ce doped samples, 1CFT, 5CFT, and 10CFT, are calculated to be 1.530, 1.478, 1.441, and 1.453 eV, respectively, as shown in Appendix A. The sample of 5CFT has the smallest band gap, and the FT sample has the largest band gap. Therefore, the Ce doping vastly increases the light absorption due to the narrowed band gap of Fe_2_O_3_.

A series of photoelectric tests further evaluated the photochemical properties of the Ce-doped Fe_2_O_3_ nanorod arrays. Obviously, the Ce-doped Fe_2_O_3_ samples, which are 1CFT, 5CFT, and 10CFT, all exhibited much higher photocurrent density than pristine FT, as shown in Figure 4a. Significantly, the photocurrent density of 5CFT (1.47 mA/cm^2^) at 1.6 V vs. RHE is 7.3 times higher than that of pristine FT (0.2 mA/cm^2^). The photocurrent density of 5CFT (0.95 mA/cm^2^) at 1.12 V vs. RHE is 6.3 times higher than that of pristine FT (0.15 mA/cm^2^). By the Ce doping, the PEC performance of Fe_2_O_3_ is highly improved. The 5CFT achieves the maximum ABPE of 1.02% at 1.39 V vs. RHE, which is 7.3 times as that of FT (0.14%), as shown in Figure 4b.

The photocurrent densities of pristine Fe_2_O_3_ and Ce-doped Fe_2_O_3_ under monochromatic light with different applied bias pressures were measured. As the applied bias increases, the photocurrent densities of these samples increase, as shown in Figure 4c,d and Appendix A. At low bias (1V vs. RHE), the photocurrent response in visible light was faintly negligible due to the high rate of recombination of electron/hole pairs. The high applied bias helps the charge separation of Fe_2_O_3_, and the effect of Ce doping in Fe_2_O_3_ can be fully exhibited, especially under visible light. The photocurrents of the Ce-doped Fe_2_O_3_ nanorods are significantly larger than that of pristine Fe_2_O_3_ in the wide range of 300 nm–600 nm. The Ce doping increases the electrical conductivity of Fe_2_O_3_ and enhances the charge separation and transfer from the bulk material to the surface. These samples’ incident photon-to-current efficiency (IPCE) was obtained, as shown in Figure 4d. The IPCE values of these samples became progressively larger as the applied voltage increased. The IPCE of 5CFT reaches a maximum value of 21% at 357 nm with bias voltage 1.6 V vs. RHE, which is 12.4 times increase to the FT sample (1.69% IPCE in the same situation).

The band gap of these samples can be evaluated by Tauc plotting [(IPCE% × hv)^1/2^ versus photon energy (hv)] under electrochemical noise mode, as shown in Figure 4e. The band gaps are 2.36 eV, 2.12 eV, 2.04 eV, and 2.17 eV of FT, 1CFT, 5CFT, and 10CFT, respectively. The substrate of TiO_2_ has a band gap of 3.2 eV. Clearly, the band gap of Fe_2_O_3_ was narrowed by Ce doping which highly enhances the visible light absorption of Fe_2_O_3_. Moreover, the Ce-doped Fe_2_O_3_ also behaved with superior photostability, as shown in Figure 4f.

The Electrochemical impedance spectroscopy (EIS) results are shown in Appendix A, where the reduction in semicircular response with frequency change can be seen. The inset shows a modified Randles circuit [41] for the EIS. The fitting parameters for each fitted component are given in Appendix A. The EIS curves of all samples show a standard semicircle, and the resistance values of the Ce-doped samples are all significantly smaller than the undoped FT, with the 5CFT sample being the smallest, demonstrating that the Ce doping enhances the electron transfer efficiency of Fe_2_O_3_ and reduces the resistance values of the samples. In addition, all the curves in the Nyquist plot pass through the origin of the coordinate axis, indicating that the resistance of the solution can be neglected, and the value of Rs is much smaller than Rp, as can be seen from the fitted circuit values.

The Mott–Schottky measurements were conducted to determine the flat band potentials of these samples, as shown in Figure 4g. Using the Mott–Schottky equation, the flat band potentials can be fitted, which are 0.01 V, 0.42 V, 0.58 V, and 0.49 V vs. RHE of FT, 1CFT, 5CFT, and 10CFT, respectively. For n-type semiconductors, the flat-band potential is positive 0.1–0.3 eV compared to the conduction band potential; for p-type semiconductors, the flat-band potential is negative 0.1–0.3 eV compared to the valence band potential. The flat band potential is infinitely near the bottom of the conduction band (CB) of the semiconductor materials, which can indirectly indicate the valence band position of the material. Although a slight positive shift shifts the CB of the Ce-doped sample compared to the FT, it still crosses the reduction potential invited and retains the ability to produce hydrogen, and the Ce-doped samples have much higher carrier concentration than pristine FT, implying high charge transfer efficiency in Ce-doped Fe_2_O_3,_ as illustrated in Appendix A. Therefore, the Ce ions doping highly promote the photocatalytic performance of Fe_2_O_3_ by facilitating the charge transfer and inhibiting the recombination of electron/hole pairs.

The H_2_ production performance of Fe_2_O_3_ samples was also explored under simulated solar light, as shown in Figure 4h. The H_2_ production of all samples increased cumulatively over time. The Ce-doped Fe_2_O_3_ showed significantly higher H_2_ production than pristine Fe_2_O_3_. After 6 h of the PEC reaction under 4.4 times AM1.5G illumination, the water-splitting H_2_ production of 5CFT achieved 126 μmol/cm^2^, which was 32 times higher than that of FT (which was 3.96 μmol/cm^2^).

### 3.2. Photoelectrochemical Mechanism for Water Splitting under Solar Light

Based on the above, the PEC mechanism of H_2_ production from water splitting was proposed, as shown in Figure 5. Without the application of external bias (left part of Figure 5a), the photogenerated electrons generated by the Fe_2_O_3_ require considerable energy to cross the potential barrier and transfer to the electrode surface. Under the applied bias, see the right part of Figure 5a, the Fermi energy level of the Fe_2_O_3_ becomes smaller, and a new Fermi energy level is formed with the TiO_2_. The conduction and valence band edges are elevated. Thus, the conduction band position of the Fe_2_O_3_ is rendered more negatively. The charge transfer of photogenerated electrons from Fe_2_O_3_ to TiO_2_ can be facilitated. Therefore, the electron and hole complexation rates are significantly reduced with the help of applied bias, and the photocatalytic performance of Fe_2_O_3_ samples was dramatically improved, as shown in Figure 5a. The band structures of pristine Fe_2_O_3_ and Ce-doped Fe_2_O_3_ nanorod arrays are illustrated in Figure 5b. A smaller band gap reduces the energy barrier that must be crossed to generate photogenerated carriers, resulting in more photogenerated carriers entering the outer circuit to participate in the redox reaction, thus enhancing the photoelectrochemical performance of the sample.

## 4. Materials and Methods

### 4.1. Materials and Synthesis Methods

Preparation of Fe_2_O_3_ nanorod arrays: The Fe_2_O_3_ nanorod arrays were prepared using a previously reported hydrothermal method [42]. The cut titanium pieces (resistance value 1–3 Ω, thickness 0.3 mm, purity 99.99%) are sonicated by acetone (C_3_H_6_O, AR ≥ 99.5%, Sinopharm Chemical Reagent Co., Ltd., Shanghai, China), deionized water (prepared by a laboratory ultrapure water machine, GZY-P10-W, Kertone Water Treatmengt Co., Ltd., Changsha, China), and ethanol absolute (C_2_H_6_O, AR ≥ 99.7%, Sinopharm Chemical Reagent Co., Ltd., Shanghai, China) for 30 min and subsequently dried under a nitrogen stream. A total of 4 mL mixture of 1 M NaNO_3_ (AR ≥ 99.0%, Sinopharm Chemical Reagent Co., Ltd., Shanghai, China) and 0.15 M FeCl_3_·6H_2_O (AR ≥ 99.0%, General-Reagent, Sinopharm Chemical Reagent Co., Ltd., Shanghai, China) was stirred for 2 h, the pH was adjusted to 1.5 by HCl (37%, Sinopharm Chemical Reagent Co., Ltd., Shanghai, China) and transfer to a Teflon vessel, meanwhile the treated titanium pieces were put together in. After holding at 100 °C for 5 h, a yellowish FeOOH film was formed on the surface of the titanium sheet and rinsed using deionized water and annealed at 550 °C for 2 h in a muffle furnace with a ramp rate of 2 °C/min, the film turns reddish brown, and the iron oxide film is successfully prepared.

Preparation of Ce-doped Fe_2_O_3_ nanorod arrays: By adding 0.01 M Ce(NO_3_)_3_ (AR ≥ 99.99%, Damas-beta) to the above Teflon vessel and sonicating for 5 min, and otherwise the same as above, we have produced an array of Ce doped iron oxide nanorods of various contents. According to the different feeding ratios, the corresponding samples are named: FT (0 μL/4 mL), 1CFT (6 μL/4 mL), 5CFT (30 μL/4 mL), and 10CFT (60 μL/4 mL). A schematic diagram of the hematite photoanodes preparation procedure is shown in Figure 6. Inductively Coupled Plasma Emission Spectrometry (ICP) and Inductively Coupled Plasma-Mass Spectrometry (ICP-MS) tests were conducted to determine the Ce content in 1CFT, 5CFT, and 10CFT as shown in Appendix A.

### 4.2. Photoelectrochemical (PEC) Testing

The Electrochemical workstation (CHI760e), a high uniformity integrated Xenon light source (PLS-FX300HU, Beijing Perfectlight, Beijing, China) with an AM 1.5G filter (100 mW/cm^2^) and three-electrode side window electrolytic cell forming the PEC reaction system for testing linear sweep voltammetry (LSV) and chronoamperometry (I-t). An LSV measurement is typically carried out over a potential range of −0.6 to 0.6 V (vs. Ag/AgCl), and the scanning speed is 5 mV s^−1^ with the light cut off by a 5 s^−1^ shutter. Chronoamperometry (I-t) was performed under alternating illumination at 0.8 V or 0.6 V (vs. Ag/AgCl). Three-electrode electrolytic cells using a platinum sheet as a counter electrode, Ag/AgCl (saturated KCl) as a reference electrode, and 1 M KOH (pH = 13.6) as electrolyte. During the test, the working electrode is guaranteed to have an illuminated area of 1 cm^2^ and is completely submerged in the electrolyte. The calculated voltage can be changed to a reversible H_2_ electrode (RHE) scale by the Nernst equation:(1)$$E_{RHE}=E_{Ag/AgCl}+0.0592\times pH+E_{Ag/AgCl}^{0}$$
where, $E_{Ag/AgCl}^{0}$ = 0.1976 V vs. *Ag/AgCl* at room temperature. The photovoltaic conversion efficiency (ABPE) with external bias can be calculated from the following equation:(2)$$ABPE\left(\%\right)=\frac{J\times \left(1.23-V\right)}{I_{0}}\times 100\%$$
where *J* is the photocurrent density, *V* is the bias voltage applied between the working electrode and reference electrode, and *I*_0_ is the intensity of the incident light (*I*_0_ = 100 mW/cm^2^).

The incident photon-to-current efficiency (IPCE) test system is composed of a 300 W xenon lamp light source (PLS-SXE300D, Beijing Perfectlight), a grating monochromator (7ISU, Shaifan) with filters to remove higher order diffraction and an electrochemical workstation (CS350H, CorrTest), which is calculated by the following equation:(3)$$IPCE\left(\%\right)=\frac{1240\times J}{\lambda \times I_{0}}\times 100\%$$
where *J* is the photocurrent density measured at a specific wavelength, *λ* is the specific wavelength, and *I*_0_ is the intensity of the incident light.

Electrochemical impedance spectroscopy (EIS) and Mott–Schottky (M-S) curves using the Admiral Squidstat Plus electrochemical workstation. The fitted circuit can be obtained from the EIS curve, where the CPE element is calculated. The impedance of the CPE in an AC circuit is:(4)$$CPE=\sigma \omega^{-m}\left[\cos\left(\frac{m\pi }{2}\right)-jsin\left(\frac{m\pi }{2}\right)\right]$$
where *σ* is the prefactor of the CPE, *ω* is the angular frequency, *m* is the CPE index (0 ≤ *m* ≤ 1), and *j* is an imaginary number $\left(j=\sqrt{-1}\right)$; if *m* = 1, then the CPE denotes the ideal capacitor C. The electrochemical analyzer works over a frequency range of 0.01 to 100,000 Hz, with voltage increments of 0.005 V and an AC amplitude of 10 mV. The working electrodes were tested at 500 Hz, 1000 Hz, 1500 Hz, 2000 Hz, and 2500 Hz. The fitted flat-band potentials allow the carrier concentration (***N_D_***) [43] to be calculated, which is calculated by the following equation:(5)$$N_{D}=\frac{2}{ke\varepsilon \varepsilon_{0}A^{2}}$$
where *k* is the slope of the fitted tangent line, *e* is the single electron charge, *ε* is the relative permittivity, *ε*_0_ is the vacuum permittivity, and *A*^2^ is the sample’s surface area. Photoelectric coupled H_2_ production tests are performed in an automated online trace gas analysis system (Labsolar-6A, Beijing Perfectlight, Beijing, China) and a gas chromatograph (F9790, Fuli Instruments, Wenling, Zhejiang, China) with an external electrochemical workstation (Squidstat Plus, Admiral Instruments, Tempe, AZ, USA) providing bias voltage. The optical power of the xenon lamp (Microsolar 300, Beijing Perfectlight Technology Co., Ltd. Beijing, China) has been tested almost equal to 4.4 times of AM 1.5G. Argon (Ar) was utilized as the carrier gas at a flow rate of 30 mL/min. Electrochemical impedance spectroscopy (EIS) and Mott–Schottky (M-S) curves were also analyzed at this workstation.

### 4.3. Materials Characterization

A Scanning Electron Microscope (SEM, Apreo S LoV ac, Thermo Fisher Scientific, Waltham, MA, USA) with an operating voltage of 10 kV was used to observe the morphology of the samples. X-ray diffraction (XRD) data were collected using a Miniflex 600 X-ray diffractometer, with CuK radiation and a measurement range of 20–80°. Transmission Electron Microscopy (TEM), High-Resolution Transmission Electron Microscopy (HRTEM), and Selected Area Electron Diffraction (SAED) were carried out on a TF20 (FEI) operating at 200 kV, Waltham, MA, USA. X-ray photoelectron spectroscopy (XPS, Thermo Scientific K-Alpha, Waltham, MA, USA) was performed to characterize the atomic composition and state at the surface of the samples, using Al Ka rays as the excitation source. UV-vis absorption spectra were gathered by a Cary 5000 spectrophotometer (Agilent Technologies, Santa Clara, CA, USA). The elemental content of Fe was determined using inductively coupled plasma emission spectrometry (ICP; UI. TIMA2, HoribaJobin Yvon S.A.S), while the content of Ce, Ti was determined using Inductively coupled plasma-Mass Spectrometry (ICP-MS; AGILENT 8800, Agilent Technologies, Santa Clara, CA, USA).

## 5. Conclusions

In conclusion, the ordered nanorod arrays of Ce-doped Fe_2_O_3_ were prepared by a simple one-step hydrothermal method for the photocatalytic H_2_ production from H_2_O. SEM and TEM images show that Fe_2_O_3_ nanorod arrays consisted of 5–30 nm nanopores with mesoporous structures. With Ce doping, the band gap structure of Fe_2_O_3_ was successfully modulated and narrowed with enhanced light absorption. Moreover, both the charge transfer and separation of photogenerated electron/hole pairs in Fe_2_O_3_ are highly improved by Ce doping due to their superior electrical conductivity. With the help of applied voltage, the doping effect of Ce doping can be further enhanced. Therefore, the Ce doped samples, 5CFT, realized a high photocurrent density of 1.47 mA/cm^2^ at 1.6 V vs. RHE, which is 7.3 times higher than that of pristine FT (0.2 mA/cm^2^). The IPCE of 5CFT reaches a maximum value of 21% at 357 nm with bias voltage 1.6 V vs. RHE, which is 12.4 times increase to the FT sample (1.69% IPCE in the same situation). Owing to the optimized band structure and enhanced PEC performance, the H_2_ production of Fe_2_O_3_ was highly improved by Ce doping to achieve an evolution rate of 21 μmol/cm^2^/h.

## Figures and Tables

**Figure 1 molecules-27-09050-f001:**
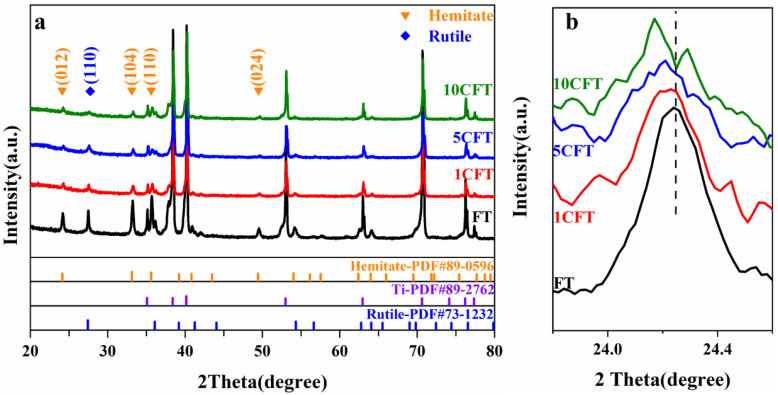
(**a**) XRD patterns of Fe_2_O_3_ (FT) and Ce-doped Fe_2_O_3_ (1CFT, 5CFT, 10CFT); (**b**) 23.5–25.0° Refined spectrum of (**a**).

**Figure 2 molecules-27-09050-f002:**
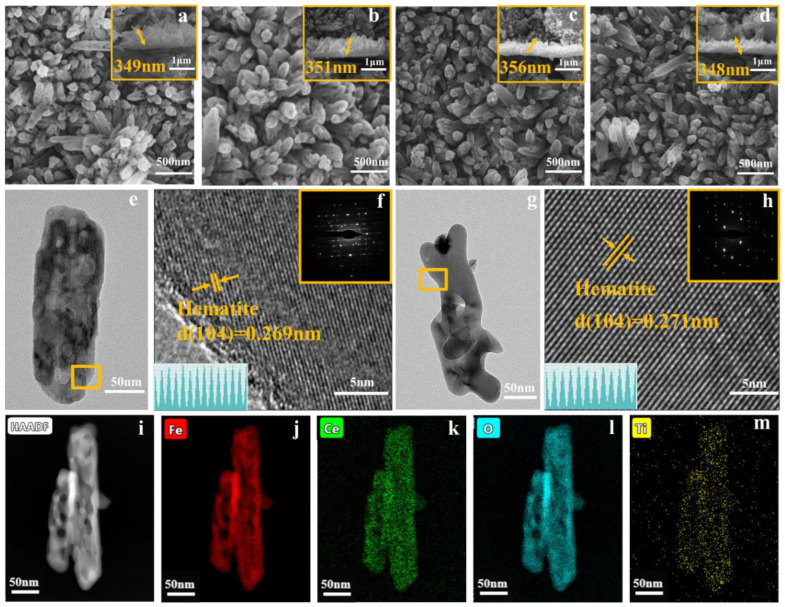
SEM and Side view of FT (**a**), 1CFT (**b**), 5CFT (**c**) and 10CFT (**d**); TEM of FT (**e**), 5CFT (**g**); HRTEM of FT (**f**), 5CFT (**h**) (Inset is SAED); HAADF (**i**) and EDS Mapping maps (**j**–**m**) of 5CFT.

**Figure 3 molecules-27-09050-f003:**
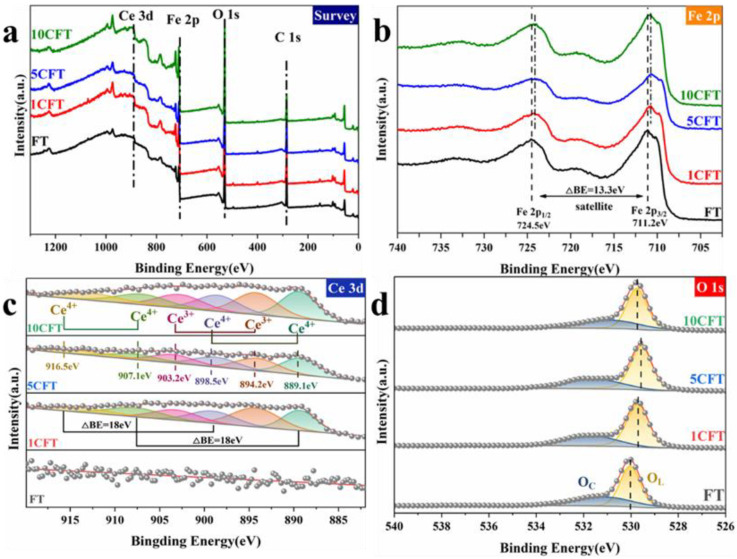
XPS spectra of Fe_2_O_3_ (FT) and Ce-doped Fe_2_O_3_ (1CFT, 5CFT, 10CFT); (**a**) the survey spectra, (**b**) Fe 2p, (**c**) Ce 3d, (**d**) O 1s.

**Figure 4 molecules-27-09050-f004:**
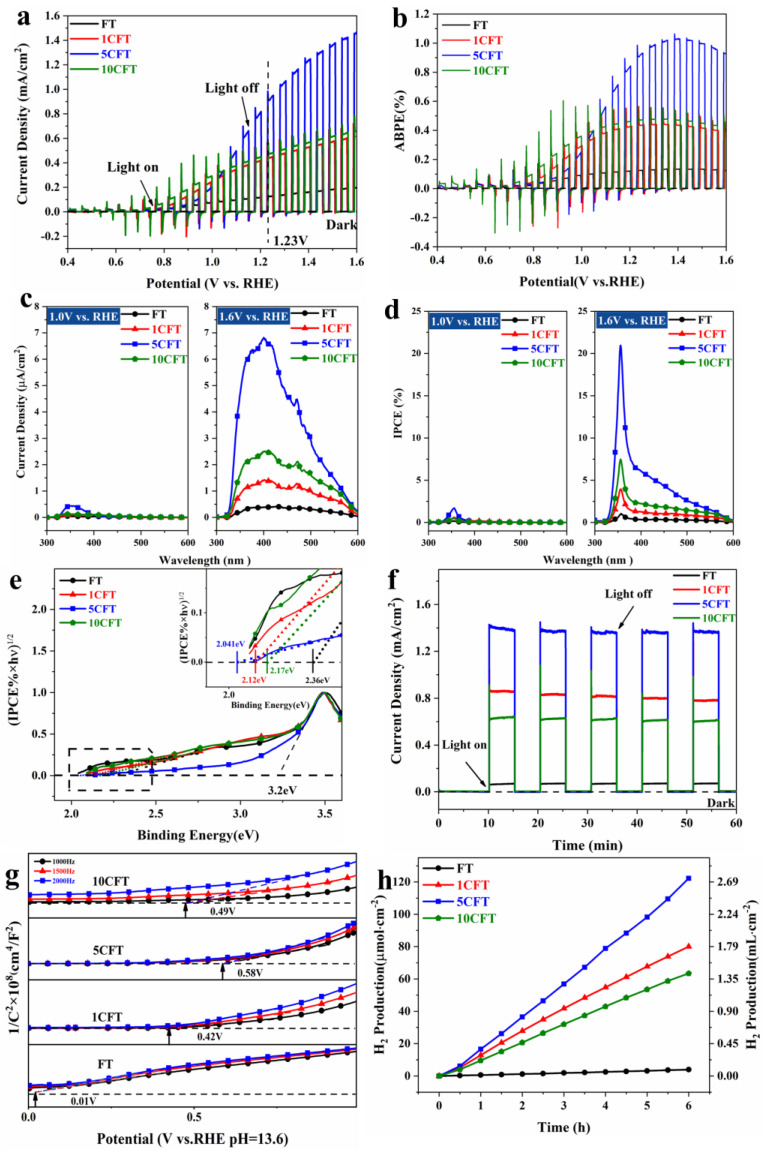
Photoelectrochemical performance of Fe_2_O_3_ (FT) and Ce-doped Fe_2_O_3_ (1CFT, 5CFT, 10CFT); (**a**) linear sweep voltammetry (**b**) ABPE (**c**) Photocurrent density versus the monochromatic light (**d**) IPCE values (**e**) Bandgap of samples (**f**) chronoamperometry (**g**) Mott–Schottky curves (**h**) H_2_ evolution properties of the samples at UV-Vis and 1.6 V vs. RHE. The light intensity received is equivalent to 4.4 suns.

**Figure 5 molecules-27-09050-f005:**
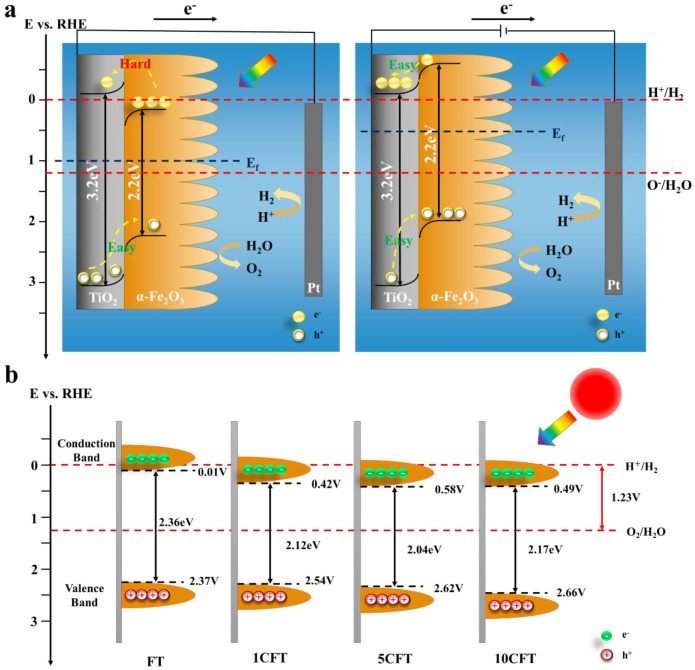
(**a**) The charge transfer mechanism of the PEC water splitting and (**b**) the schematic diagram of the band gap of Fe_2_O_3_ (FT) and Ce-doped Fe_2_O_3_ (1CFT, 5CFT, 10CFT).

**Figure 6 molecules-27-09050-f006:**
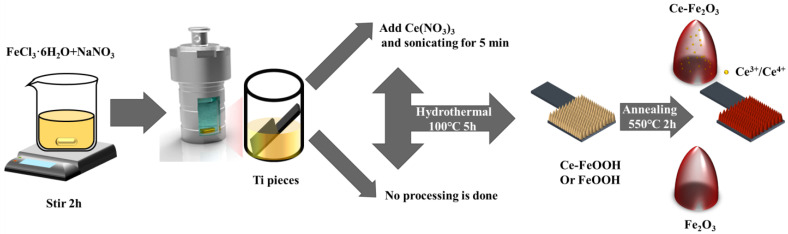
Schematic diagram of hematite photoanodes preparation procedures.

## Data Availability

Not applicable.

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
