# Peer review of "Cerium-Doped Iron Oxide Nanorod Arrays for Photoelectrochemical Water Splitting"

_molecules, 2022, doi:10.3390/molecules27249050_

Round 1

Reviewer 1 Report

* Structure of  paper should be modified. 

* Order should be 

Introduction, Material  synthesis, Chracaterization ,  Results & discussion.

* In synthesis portion , synthesis of nanoparticle should be written in detail.

* In this paper many times short form is used. Kindly write the name of short form in the bracket.

Reviewer 2 Report

Manuscript deals with hydrothermal obtaining of the Ce-doped hematite supported on titanium for water splitting.

However, some issues have to be addressed.

Page 2, line 66- Results are presented in the aim of the paper. “The optical properties and photocatalytic activity of Fe2O3 were remarkably improved by Ce doping” and “Therefore, the Ce doped samples realized high photocurrent density and IPCE values. The PEC performance for H2 production of Fe2O3 was highly improved by Ce doping.”

Page 2- Sample names should be used prior to their abbreviation.

Page 5- It is hard to depict any XPS peaks of the Ce 3d in Figure 3c (Fig. 5c is written in line 123, page 5?!)

Page 11- A more detailed material characterization procedure is required.

Comparison on H2 production with other ceria- doped materials will be helpful for proving ceria effect.

Reviewer 3 Report

In the present manuscript, authors have reported the synthesis and characterization of Cerium-doped iron oxide nanorod arrays for photoelectrochemical water splitting. The subject is interesting and results are discussed scientifically. The manuscript may be accepted after few minor changes.

 1) Authors have used many abbreviations which are either defined in later part of manuscript or not defined al all e.g PEC and RHE in the abstract; IPCE at page 2 line 71. Authors must define an abbreviation where it is used for the first time and then strictly adhere to that for clarity in text. Although few abbreviation are common e.g. RHE but even then it is better to define them in the manuscript for convenience of readers.

2) In abstract, Line 26: sentence “The applied voltage…. effect of Ce in Fe2O3.” is not clear. It needs to be revised.

3) There are some typos/grammatical mistakes in the manuscript e.g Page 1 Line 38: replace “consumes” with “consume” ; page 1 Line 40: replace “environmentally” with “environmental” ; page 2 line 45: Metal oxide semiconductors (MOS) “have”…. etc. Authors should carefully revise the manuscript to remove such mistakes.  

4) Captions of Figures S2, S3, S4 and Table S2 are not comprehensive. Make them comprehensive like caption of Figure S5.

5) Use different symbols along with color scheme to represent different sample in Figures S3 and S5. It will help the readers to understand the results even in black & white print.
